# Sex, age, and racial/ethnic disparities in hyperuricemia prevalence and risk factors among U.S. adults: An analysis of NHANES 2007–2018 data

Yadan Zou[1], Lina Zhang[1], Jing Xu[1], Ji Li[1], Jing Zhang[1], Ting Long[1], Ruohan Yu[1], Yanfeng Zhang[1], Zhongxing Zhao[2], Sheng-Guang Li[1]*

1 Department of Rheumatology and Immunology, Peking University International Hospital, Beijing, China,
2 Shenyang Medical College, ShenYang, Liao Ning, China

☯ These authors contributed equally to this work
* lishengguang@vip.sina.com

## Abstract

### Background

Hyperuricemia is a metabolic disorder linked to gout, kidney disease, and cardio-vascular complications. Understanding its prevalence and risk factors across demographic groups is crucial for effective prevention and management.

### Objectives

To evaluate the prevalence of hyperuricemia among U.S. adults by sex, age, and racial/ethnic groups, and identify common and sex-specific risk factors.

### Methods

Data from 34,144 U.S. adults aged ≥20 years, obtained from the National Health and Nutrition Examination Survey (NHANES) 2007–2018, were analyzed. Hyperuricemia was defined as serum uric acid levels >7.0 mg/dL in males and >6.0 mg/dL in females. Prevalence estimates were calculated, and multivariate logistic regression models identified risk factors, adjusting for confounders such as body mass index (BMI), alcohol consumption, hypertension, renal function, and other variables. A sensitivity analysis excluding participants with a history of gout was conducted to evaluate the robustness of identified associations.

### Results

The overall prevalence of hyperuricemia was significantly higher in males (21.1%) compared to females (17.1%, $P < 0.001$). Females surpassed males in both prevalence and absolute numbers after age 50–59; by age ≥80, the number of female

**Data availability statement:** This study is based on publicly available data from the U.S. National Health and Nutrition Examination Survey (NHANES). The dataset is maintained by the National Center for Health Statistics (NCHS), Centers for Disease Control and Prevention (CDC). The data can be freely accessed at the NHANES website (https://www.cdc.gov/nchs/nhanes/). No special access privileges were required to obtain these data; other researchers can access the datasets in the same manner as the authors by selecting the relevant survey cycles (2007–2018) and downloading the data files provided.

**Funding:** The author(s) received no specific funding for this work.

**Competing interests:** The authors have declared that no competing interests exist.

cases was more than twice that of males. Non-Hispanic Black adults had the highest prevalence (23.9% in males, 23.4% in females). Key risk factors for both sexes included obesity (OR = 3.91 in males; OR = 4.76 in females), hypertension (OR = 1.65 in males; OR = 2.09 in females), and impaired renal function (eGFR < 30 mL/min/1.73 m²: OR = 3.72 in males; OR = 15.37 in females). Alcohol consumption was positively associated with hyperuricemia in males (OR = 1.25), but not significantly so in females. Diabetes showed opposite associations: protective in males (OR = 0.72) but a risk factor in females (OR = 1.22). Medication use exhibited expected directional effects: diuretic use was associated with significantly increased risk of hyperuricemia (OR = 2.67 in males; OR = 2.55 in females), while urate-lowering therapy was associated with reduced risk (OR = 0.57 in males; OR = 0.55 in females).

## Conclusions

Hyperuricemia remains highly prevalent in the U.S., with notable disparities by sex, age, and race/ethnicity. Older women bear a particularly high burden, partly due to obesity, renal dysfunction, and diuretic use. Incorporating medication use into analyses strengthens the evidence for sex-specific risk profiles. These findings highlight the importance of considering targeted screening and prevention strategies in specific high-risk groups, such as older women and patients receiving diuretics.

## Introduction

Hyperuricemia, defined by elevated serum uric acid (SUA) levels, is a metabolic disorder closely linked to gout and multiple chronic comorbidities such as chronic kidney disease (CKD), hypertension, diabetes mellitus, and cardiovascular diseases (CVD) [1–4]. The global prevalence of hyperuricemia has been increasing, paralleling the rise in obesity, hypertension, and other lifestyle-related disorders [5]. Given its widespread occurrence and associated health risks, understanding demographic variations and identifying key risk factors for hyperuricemia is essential for developing effective prevention and management strategies.

Sex differences in hyperuricemia prevalence have been consistently reported, with men typically exhibiting higher SUA levels compared to women. This disparity is primarily attributed to hormonal influences, particularly the uricosuria effect of estrogen in premenopausal women, facilitating renal excretion of uric acid, thereby maintaining lower SUA levels in this population [6]. After menopause, the protective effect of estrogen diminishes, leading to increased hyperuricemia prevalence among women [7]. However, comprehensive evaluations comparing both the prevalence and the absolute number of hyperuricemia cases between sexes across various age groups are limited, especially within nationally representative populations.

Age is another critical factor influencing hyperuricemia prevalence, reflecting a complex interplay between hormonal changes, metabolic shifts, and progressive renal impairment associated with aging [8]. While increased age is generally linked to

elevated SUA levels, it remains uncertain whether hyperuricemia prevalence and absolute patient numbers among older women eventually exceed those among older men. Clarifying these trends is important for tailoring clinical interventions and targeting preventive measures to demographic groups particularly vulnerable to hyperuricemia-related health issues.

Racial and ethnic disparities in hyperuricemia prevalence have also been observed, potentially driven by genetic pre-dispositions, socioeconomic status, dietary practices, and disparities in healthcare access. Previous studies suggest that Non-Hispanic Black and Hispanic populations may experience a higher prevalence of hyperuricemia compared to other racial and ethnic groups. Nevertheless, comprehensive analyses accounting simultaneously for sex and age differences within these groups remain scarce.

In this study, we analysed data from the National Health and Nutrition Examination Survey (NHANES) 2007–2018 cycles, representing the U.S. adult population, to provide updated hyperuricemia prevalence estimates. We specifically examined variations in hyperuricemia prevalence by sex, age, and racial/ethnic subgroups and explored both common and sex-specific risk factors. We also seek to identify common and sex-specific risk factors associated with hyperuricemia, with a particular focus on examining whether the prevalence and actual number of women with hyperuricemia surpass those of men with increasing age. By addressing these objectives, we hope to fill gaps in the current understanding of hyperuricemia's epidemiology in the United States.

Our findings are intended to inform targeted screening practices, guide personalized preventive strategies, and support broader public health initiatives aimed at reducing the clinical and societal burden of hyperuricemia and its associated conditions.

## Methods

### Ethics statement

To ensure ethical compliance, the protocols undergo review by the NCHS Ethics Review Committee, with informed consent obtained from all participants. Data from NHANES is publicly available through their official website (http://www.cdc.gov/nchs/nhanes.htm), accessed on March 1, 2022. Authors had no access to information that could identify individual participants during or after data collection.

### Study design and population

This cross-sectional study utilized data from the National Health and Nutrition Examination Survey (NHANES) cycles 2007–2018. NHANES employs a multistage, stratified probability sampling design to assess health and nutritional status among the civilian, non-institutionalized U.S. population.

Participants aged ≥ 20 years who had relevant demographics, and health-related questionnaires were included. Initially, the dataset comprised 34770 participants. Among these, missing SUA values (3,405) were addressed through Multiple Imputation by Chained Equations (MICE) to minimize potential biases related to missing data (Fig 1).

### Data collection and variable definitions

**Hyperuricemia Identification.** Hyperuricemia was defined as a serum uric acid (SUA) concentration >7.0 mg/dL (>416 µmol/L) in males and >6.0 mg/dL (>357 µmol/L) in females. These thresholds were based on the 2018 European Alliance of Associations for Rheumatology (EULAR) evidence-based recommendations for the diagnosis of gout [9] and the 2020 American College of Rheumatology (ACR) guideline for the management of gout [10]. This definition reflects the approximate physiological saturation point of monosodium urate under normal temperature and pH conditions, while also accounting for sex-specific differences in uric acid metabolism. Serum uric acid was measured enzymatically using a uricase-based method. Specifically, for NHANES 2007–2008 cycles, uric acid was measured using the Beckman Synchron LX20 analyzer, while from 2009–2018, measurements were performed using the Beckman Coulter UniCel

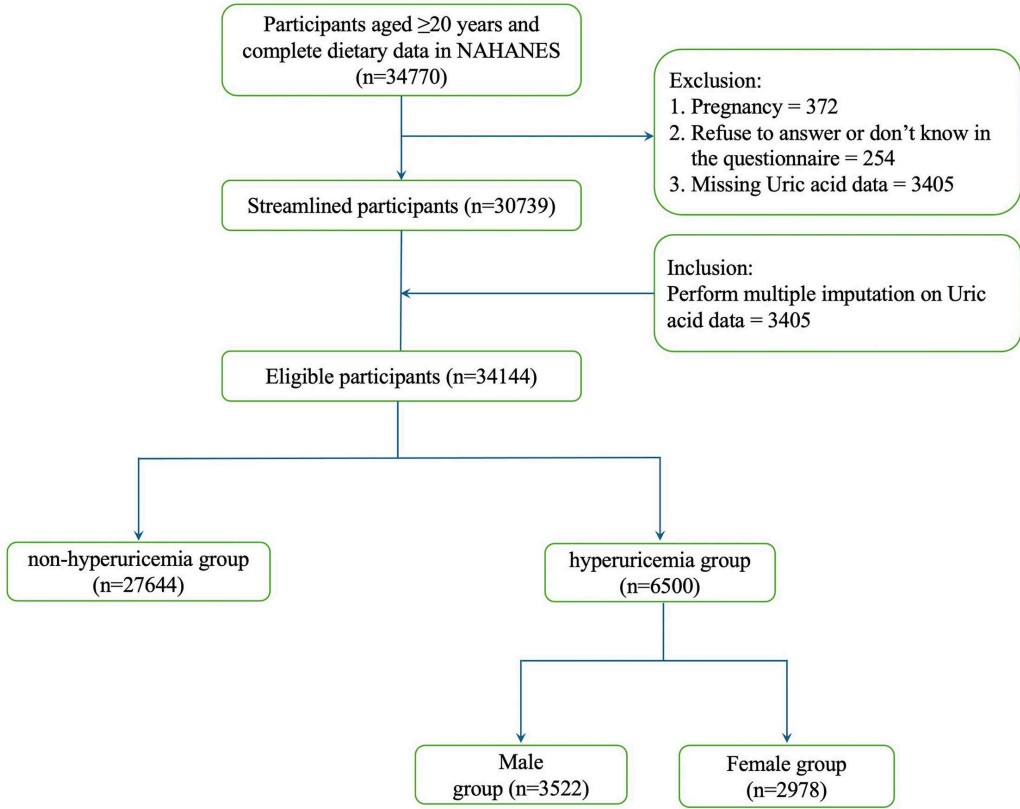

**Fig 1. Flowchart of Participant Selection from NHANES 2007–2018.**

DxC800 Synchron auto-analyzer. All laboratory measurements followed standardized NHANES protocols, including regular calibration, and were subject to rigorous quality control procedures.

**Demographic variables.** A range of potential covariates was evaluated based on the existing literature [11], encompassing age, sex, marital status, race/ethnicity, education level, family income, gout, hypercholesterolemia, hypertension, diabetes, coronary heart disease, alcohol consumption, BMI, and GFR. Age was analyzed categorically (20–39, 40–59, 60–79, ≥ 80 years). Race/ethnicity was classified as non-Hispanic White, non-Hispanic Black, Mexican American, and other race/ethnicity groups. Marital status was categorised as married or living alone. Education level was divided into less than high school, high school graduate or GED, some college or associate degree, and college graduate or above. Family income was classified into three groups based on the poverty income ratio (PIR): low (PIR ≤ 1.0), medium (PIR > 1 to 2.0), and high (PIR > 3).

Estimated glomerular filtration rate (eGFR) was calculated using the Chronic Kidney Disease Epidemiology Collaboration (CKD-EPI) creatinine equation, based on serum creatinine measured in MEC laboratory assessments.

**Clinical and anthropometric Variables.** Body mass index (BMI) was calculated as weight in kilograms divided by height in meters squared (kg/m²), based on standardized anthropometric measurements collected during the MEC physical examination. Participants were classified according to standard categories: normal (<25 kg/m²), overweight (25–29.9 kg/m²), and obese (≥30 kg/m²). Hypertension was identified by either self-report of physician diagnosis, current antihypertensive medication use, or measured blood pressure ≥140/90 mmHg during NHANES examinations. Diabetes mellitus was defined by self-reported physician diagnosis, current use of diabetes medications (oral hypoglycemic

agents or insulin), fasting plasma glucose ≥126 mg/dL, or HbA1c ≥ 6.5%.A history of gout, and coronary heart disease was determined based on self-reported physician-diagnosed conditions. Hypercholesterolemia was defined as a total serum cholesterol concentration ≥240 mg/dL [12], consistent with the National Cholesterol Education Program Adult Treatment Panel III (NCEP ATP III) criteria and widely used in NHANES-based epidemiological studies. Total cholesterol was measured from blood samples obtained in the Mobile Examination Center (MEC) and analyzed in accordance with standardized laboratory protocols.

Renal function was assessed by estimated glomerular filtration rate (eGFR), calculated using the CKD-EPI equation. Chronic kidney disease (CKD) stages were categorized based on eGFR: normal (≥90 mL/min/1.73m²), mild(60–89 mL/min/1.73m²), moderate (30–59 mL/min/1.73m²)and severe impairment (<30 mL/min/1.73m²).Alcohol drinking status was determined using the survey question "In any 1 year, have you had at least 12 drinks of any type of alcoholic beverage?" Participants who answered "yes" were classified as alcohol drinkers.

## Statistical analysis

**Descriptive Analysis:** Summarized demographic characteristics and hyperuricemia prevalence across subgroups using chi-square tests for categorical variables and t-tests for continuous variables.

**Logistic Regression Analysis:** Conducted univariate and multivariate logistic regression to identify risk factors for hyperuricemia [13]. Covariates included age, sex, race/ethnicity, BMI, diabetes, hypertension, CKD, alcohol intake, smoking status, and income. Odds ratios (ORs) and 95% confidence intervals (CIs) were reported.

Due to potential reverse causality, a sensitivity analysis excluding participants with a history of gout was also conducted.

all models incorporated the NHANES complex survey design. Specifically, we applied the 2-year MEC examination weights (WTMEC2YR), divided by six to create a 12-year pooled weight, and specified strata (SDMVSTRA) and primary sampling units (SDMVPSU). Analyses were conducted using R (version 4.2.2) with the *survey* package, ensuring nationally representative estimates. In addition, use of diuretics (e.g., furosemide) and urate-lowering therapy (ULT, e.g., allopurinol, febuxostat) was included as binary covariates, derived from NHANES medication data files.

**Stratified Analysis:** To explore demographic disparities, stratified analyses were performed by age groups (20–29, 30–39,40–49, 50–59,60–69,70–79, ≥ 80 years), sex, and race/ethnicity.

**Handling of missing data.** Missing serum uric acid values were addressed using Multiple Imputation by Chained Equations (MICE) [14], specifying five imputations to create complete datasets. MICE was chosen for its robustness in handling missing data across diverse demographic variables and its ability to reduce bias in the analysis.

## Results

### Characteristics of the study population

The final sample comprised 34,144 U.S. adults from the NHANES 2007–2018 cycles. Among them, 6,500 individuals were identified with hyperuricemia, including 3,522 males (54.18%) and 2,978 females (45.82%, Fig 1). The sample encompassed diverse ages, sexes, and racial/ethnic backgrounds.

### Prevalence of hyperuricemia by sex, age, and race/ethnicity

**Sex differences.** Males had a significantly higher overall prevalence of hyperuricemia (21.1%) compared to females (17.1%) (p < 0.001), affecting approximately 23.79 million males and 20.43 million females in the U.S. (Table 1).

**Age trends.** In males, the prevalence of hyperuricemia remained consistently high from age 20 onwards, with minimal variation across age groups. Specifically, the prevalence was 19.7% in the 20–29 age group and remained relatively stable in older groups.

**Table 1. Prevalence of hyperuricemia, estimated number of U.S. adults affected, and mean serum uric acid levels by sex, age group, and race/ethnicity (NHANES 2007–2018).**

| | Prevalence of hyperuricemia, % (95% CI) | | | No. of US adults with hyperuricemia, millions | | | Serum urate level, mean (95% CI) mg/dL | | |
|---|---|---|---|---|---|---|---|---|---|
| | Male | Female | Both | Male | Female | Both | Male | Female | Both |
| Total population | 21.1 (20.5, 21.7) | 17.1 (16.5, 17.6) | 19.0 (18.6, 19.5) | 23.79 | 20.43 | 44.12 | 6.02 (6.00, 6.04) | 4.87 (4.85, 4.89) | 5.43 (5.42, 5.44) |
| **Age** | | | | | | | | | |
| 20-29 | 19.7 (18.2, 21.2) | 8.7 (7.7, 9.8) | 14.3 (13.4, 15.2) | 4.42 | 1.87 | 6.28 | 5.98 (5.93, 6.02) | 4.52 (4.48, 4.56) | 5.26 (5.23, 5.29) |
| 30-39 | 20.7 (19.2, 22.2) | 8.8 (7.7, 9.8) | 14.6 (13.7, 15.6) | 4.29 | 1.81 | 6.03 | 6.05 (6.00, 6.10) | 4.50 (4.46, 4.54) | 5.27 (5.23, 5.30) |
| 40-49 | 20.3 (18.8, 21.8) | 10.5 (9.4, 11.6) | 15.1 (14.1, 16.0) | 4.27 | 2.24 | 6.40 | 6.02 (5.97, 6.07) | 4.57 (4.53, 4.62) | 5.25 (5.22, 5.28) |
| 50-59 | 19.7 (18.2, 21.2) | 18.3 (16.8, 19.7) | 19.0 (18.0, 20.0) | 4.09 | 3.98 | 8.08 | 5.93 (5.88, 5.98) | 4.96 (4.91, 5.01) | 5.44 (5.40, 5.47) |
| 60-69 | 21.7 (20.2, 23.2) | 24.2 (22.7, 25.8) | 23.0 (21.9, 24.1) | 3.30 | 4.06 | 7.35 | 6.05 (6.00, 6.10) | 5.17 (5.12, 5.22) | 5.61 (5.57, 5.64) |
| 70-79 | 23.5 (21.6, 25.5) | 30.3 (28.2, 32.4) | 27.0 (25.5, 28.4) | 1.95 | 3.06 | 4.97 | 6.09 (6.02, 6.15) | 5.43 (5.36, 5.49) | 5.75 (5.71, 5.80) |
| ≥80 | 25.0 (22.4, 27.6) | 31.8 (29.2, 34.4) | 28.7 (26.8, 30.5) | 1.09 | 2.35 | 3.37 | 6.04 (5.95, 6.13) | 5.45 (5.37, 5.54) | 5.73 (5.67, 5.79) |
| **Race/ethnicity, n (%)** | | | | | | | | | |
| Non-Hispanic white | 21.6 (20.6, 22.6) | 17.9 (17.0, 18.8) | 19.8 (19.1, 20.4) | — | — | — | 6.04 (6.01, 6.07) | 4.90 (4.87, 4.93) | 5.47 (5.45, 5.49) |
| Non-Hispanic black | 23.9 (22.5, 25.3) | 23.4 (22.0, 24.7) | 23.6 (22.6, 24.6) | — | — | — | 6.12 (6.07, 6.17) | 5.11 (5.06, 5.16) | 5.60 (5.57, 5.64) |
| Mexican American | 16.7 (15.2, 18.2) | 11.5 (10.2, 12.7) | 14.0 (13.1, 15.0) | — | — | — | 5.83 (5.78, 5.88) | 4.64 (4.59, 4.69) | 5.22 (5.19, 5.26) |
| Others | 20.3 (19.0, 21.6) | 13.5 (12.4, 14.5) | 16.7 (15.9, 17.5) | — | — | — | 6.01 (5.97, 6.05) | 4.73 (4.69, 4.76) | 5.33 (5.31, 5.36) |

In contrast, females exhibited a progressive increase in hyperuricemia prevalence with age after 40–49 years. The prevalence rose from 8.7% in the 20–29 age group to 30.3% in the 70–79 age group, surpassing males after the age of 50–59 (Table 1 and Fig 2A).

Analyzing the estimated absolute number of affected individuals revealed distinct patterns between sexes. For males, the number of hyperuricemia patients exceeded 4 million in each age group before 50–59 years old. After age 50–59, despite the prevalence remaining stable, the estimated number of male patients in each age group showed a rapid decline (Table 1 and Fig 2B).

Conversely, among females, the estimated number of hyperuricemia patients before the 40–49 age group was lower than that of males. After age 50–59, the number of female patients increased rapidly, eventually approaching and surpassing the number of males. Although the number of female patients began to decline after the 60–69 age group, the difference compared to males continued to widen. By age 80 and above, the number of female hyperuricemia patients was more than twice that of males (Fig 2B).

**Racial/Ethnic disparities.** Non-Hispanic Black adults had the highest prevalence of hyperuricemia (23.9% in males and 23.4% in females), followed by non-Hispanic Whites (21.6% in males and 17.9% in females). Mexican Americans had the lowest prevalence (16.7% in males and 11.5% in females, Fig 3A).

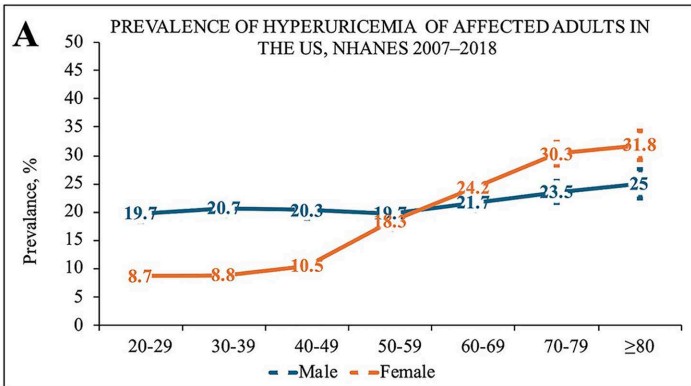
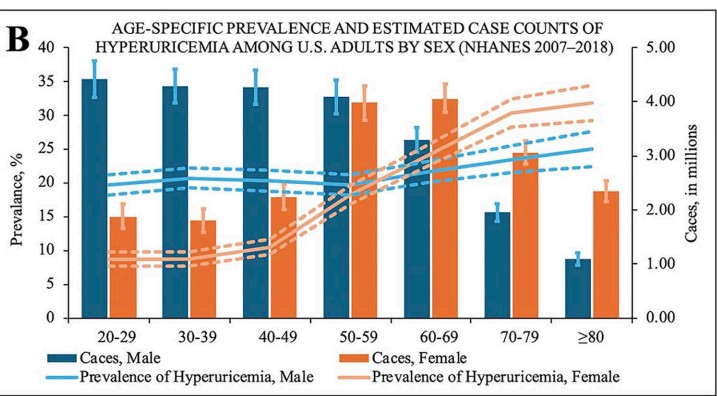

**Fig 2. (2A and 2B). Age- and Sex-Specific Patterns in Gout and Hyperuricemia Prevalence and Estimated Case Counts (NHANES 2007–2018).**

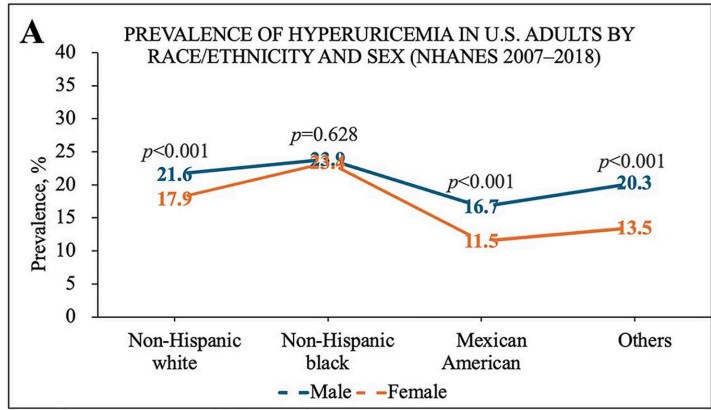
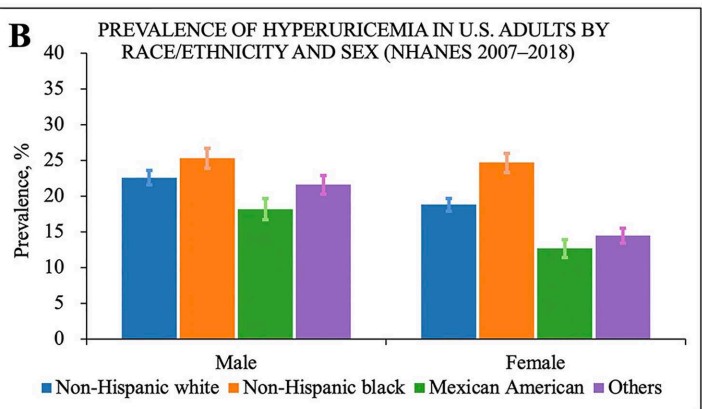

**Fig 3. (3A and 3B). Racial/Ethnic and Sex-Specific Prevalence of Hyperuricemia Among U.S. Adults (NHANES 2007–2018).**

Regarding sex differences, the prevalence in non-Hispanic Black males and females was nearly identical, showing almost no disparity (Fig 3A and 3B). In contrast, among non-Hispanic Whites, Mexican Americans, and other racial/ethnic groups, males had a significantly higher prevalence than females ($P < 0.001$).

**Risk factors for hyperuricemia.** An analysis of hyperuricemia-related factors revealed both common and distinct risk patterns between sexes. After multivariate adjustment, BMI, gout, alcohol consumption, hypertension, and decreased renal function were common risk factors for both males and females. However, the impact of each factor and the role of specific variables exhibited significant gender differences (Table 2 and Table 3).

## Common risk factors

1. **Body Mass Index (BMI):** Elevated BMI significantly increased the risk of hyperuricemia in both sexes. In the BMI ≥ 30 kg/m² group, the risk increased by 3.91-fold in males (OR = 3.91, 95% CI: 3.32–4.60, $P < 0.001$) and 4.76-fold in females (OR = 4.76, 95% CI: 3.89–5.81, $P < 0.001$).

**Table 2. Weighted Univariate and Multivariate Logistic Regression Analysis of Risk Factors for Hyperuricemia in Males (NHANES 2007–2018).**

| | Male with Hyperuricemia/n(%) | Univariate OR (95%CI) | Multivariate OR (95%CI) | Univariate p | Multivariate p |
|---|---|---|---|---|---|
| **Age** | | | | | |
| 20-29 | 546 (19.70%) | 1.00 (Referent) | 1.00 (Referent) | | |
| 30-39 | 582 (20.66%) | 1.09 (0.92, 1.28) | 0.79 (0.66, 0.94) | 0.340 | 0.011 |
| 40-49 | 540 (20.31%) | 1.02 (0.84, 1.24) | 0.60 (0.49, 0.74) | 0.840 | < 0.001 |
| 50-59 | 537 (19.74%) | 0.90 (0.75, 1.08) | 0.45 (0.35, 0.56) | 0.265 | < 0.001 |
| 60-69 | 626 (21.73%) | 1.04 (0.88, 1.23) | 0.40 (0.32, 0.50) | 0.649 | < 0.001 |
| 70-79 | 422 (23.55%) | 1.09 (0.89, 1.34) | 0.30 (0.23, 0.40) | 0.416 | < 0.001 |
| ≥80 | 269 (25.00%) | 1.20 (0.97, 1.49) | 0.28 (0.21, 0.38) | 0.092 | < 0.001 |
| **Race/ethnicity, n (%)** | | | | | |
| Non-Hispanic white | 1495 (21.61%) | 1.00 (Referent) | 1.00 (Referent) | | |
| Non-Hispanic black | 854 (23.88%) | 1.10 (0.99, 1.22) | 1.11 (0.99, 1.24) | 0.080 | 0.084 |
| Mexican American | 417 (16.69%) | 0.78 (0.65, 0.92) | 0.77 (0.64, 0.92) | 0.004 | 0.006 |
| Others | 756 (20.30%) | 0.98 (0.85, 1.13) | 1.18 (1.01, 1.39) | 0.818 | 0.046 |
| **Education** | | | | | |
| Some high school | 861 (19.84%) | 1.00 (Referent) | 1.00 (Referent) | | |
| High school or GED | 864 (21.60%) | 1.17 (1.02, 1.35) | 1.07 (0.93, 1.25) | 0.025 | 0.354 |
| Some college | 1018 (22.66%) | 1.23 (1.07, 1.41) | 1.06 (0.90, 1.23) | 0.004 | 0.502 |
| College graduate | 779 (20.05%) | 0.96 (0.82, 1.13) | 0.94 (0.77, 1.15) | 0.648 | 0.561 |
| **Marital status, n (%)** | | | | | |
| Married or living with a partner | 2229 (20.78%) | 1.00 (Referent) | 1.00 (Referent) | | |
| Living alone | 1293 (21.59%) | 1.08 (0.96, 1.21) | 0.87 (0.77, 0.98) | 0.221 | 0.026 |
| **Ratio of family income to Poverty** | | | | | |
| ≤1.0 | 703 (20.85%) | 1.00 (Referent) | 1.00 (Referent) | | |
| 1.0 to 2.0 | 954 (21.04%) | 1.09 (0.96, 1.23) | 1.04 (0.91, 1.19) | 0.176 | 0.595 |
| > 2.0 | 1865 (21.17%) | 1.11 (0.98, 1.25) | 1.07 (0.93, 1.24) | 0.096 | 0.352 |
| **Body mass index** | | | | | |
| ≤24.9 kg/m2 | 539 (11.32%) | 1.00 (Referent) | 1.00 (Referent) | | |
| 25.0 kg/m2 to 29.9 kg/m2 | 1181 (19.17%) | 2.20 (1.89, 2.57) | 2.11 (1.80, 2.48) | < 0.001 | < 0.001 |
| ≥30.0 kg/m2 | 1802 (31.09%) | 4.16 (3.59, 4.83) | 3.91 (3.32, 4.60) | < 0.001 | < 0.001 |
| **Alcohol Use** | | | | | |
| No | 843 (19.83%) | 1.00 (Referent) | 1.00 (Referent) | | |
| Yes | 2679 (21.49%) | 1.14 (1.02, 1.28) | 1.25 (1.10, 1.42) | 0.021 | 0.001 |
| **Diabetes** | | | | | |
| No | 2979 (20.71%) | 1.00 (Referent) | 1.00 (Referent) | | |
| Yes | 543 (23.24%) | 1.08 (0.93, 1.26) | 0.72 (0.60, 0.87) | 0.323 | 0.001 |
| **Hypercholesterolemia** | | | | | |
| No | 1456 (23.99%) | 1.00 (Referent) | 1.00 (Referent) | | |
| Yes | 2066 (19.40%) | 0.73 (0.66, 0.82) | 0.73 (0.65, 0.82) | < 0.001 | < 0.001 |
| **Hypertension** | | | | | |
| No | 1857 (17.29%) | 1.00 (Referent) | 1.00 (Referent) | | |
| Yes | 1665 (27.87%) | 1.82 (1.65, 2.02) | 1.65 (1.46, 1.86) | < 0.001 | < 0.001 |
| **Coronary heart disease** | | | | | |
| No | 3271 (20.77%) | 1.00 (Referent) | 1.00 (Referent) | | |
| Yes | 251 (25.98%) | 1.19 (0.92, 1.54) | 0.87 (0.63, 1.20) | 0.180 | 0.403 |

*(Continued)*

**Table 2.** (Continued)

| | Male with Hyperuricemia/n(%) | Univariate OR (95%CI) | Multivariate OR (95%CI) | Univariate p | Multivariate p |
|---|---|---|---|---|---|
| **Glomerular filtration rate (GFR)** | | | | | |
| GFR ≥ 90mL/min | 1597 (16.70%) | 1.00 (Referent) | 1.00 (Referent) | | |
| GFR 60 to 89mL/min | 1316 (23.24%) | 1.43 (1.28, 1.60) | 1.92 (1.66, 2.23) | < 0.001 | < 0.001 |
| GFR 30 to 59mL/min | 545 (40.89%) | 3.55 (2.96, 4.25) | 6.10 (4.75, 7.84) | < 0.001 | < 0.001 |
| GFR < 30mL/min | 64 (39.75%) | 2.92 (1.80, 4.74) | 3.72 (2.11, 6.54) | < 0.001 | < 0.001 |
| **Furosemide** | | | | | |
| No | 3260 (20.16%) | 1.00 (Referent) | 1.00 (Referent) | | |
| Yes | 262 (48.16%) | 3.42 (2.66, 4.40) | 2.67 (1.97, 3.61) | < 0.001 | < 0.001 |
| **Antihyperuricemic** | | | | | |
| No | 3477 (21.03%) | 1.00 (Referent) | 1.00 (Referent) | | |
| Yes | 45 (24.73%) | 1.00 (0.64, 1.56) | 0.57 (0.35, 0.95) | 0.997 | 0.034 |

Adjusted for age, race/ethnicity, education level, marriage, income, body mass index, hypertension, diabetes, renal function (eGFR), alcohol consumption, smoking status, history of gout, diuretic use, and urate-lowering therapy use.

2. **Hypertension:** Hypertension significantly increased the risk of hyperuricemia in both sexes, with a 1.65-fold increase in males (OR = 1.65, 95% CI: 1.46–1.86, P < 0.001) and a 2.09-fold increase in females (OR = 2.09, 95% CI: 1.83–2.39, P < 0.001).

3. **Decreased Renal Function (eGFR):** Decline in renal function was the strongest risk factor. Patients with eGFR < 30 mL/min/1.73 m² had an increased risk of 3.42-fold in males (OR = 3.42, 95% CI: 2.11–6.54, P < 0.001) and 15.37-fold in females (OR = 15.37, 95% CI: 9.65–24.48, P < 0.001).

4. **Alcohol Consumption:** Alcohol use was associated with increased hyperuricemia risk in males (OR = 1.25, 95% CI: 1.10–1.42, P = 0.001), but not in females (OR = 1.15, 95% CI: 0.99–1.33, P = 0.068).

5. **Diabetes Mellitus:** Diabetes showed opposite associations by sex. In males, diabetes was inversely associated with hyperuricemia (OR = 0.72, 95% CI: 0.60–0.87, P = 0.001). In contrast, in females, diabetes was positively associated (OR = 1.22, 95% CI: 1.04–1.42, P = 0.016).

6. **Medication Use:** Medication variables demonstrated expected effects. Diuretic use (e.g., furosemide) was associated with higher odds of hyperuricemia in both sexes (males: OR = 2.67, 95% CI: 1.97–3.61, P < 0.001; females: OR = 2.55, 95% CI: 1.98–3.27, P < 0.001). Urate-lowering therapy was strongly protective (males: OR = 0.57, 95% CI: 0.35–0.95, P = 0.034; females: OR = 0.55, 95% CI: 0.21–1.43, P = 0.227).

## Sex differences

**Impact of age.** **Males:** As see in Table 2 (and S1 Table), After multivariate adjustment, age showed an inverse association with hyperuricemia risk. In the ≥ 80 years group, the risk decreased significantly (OR = 0.28, 95% CI: 0.21–0.38, P < 0.001).

**Females:** Univariate analysis demonstrated that increasing age significantly elevated hyperuricemia risk. However, after multivariate adjustment in the analysis excluding participants with gout, the association was substantially attenuated and became non-significant across all older age groups (e.g., for age ≥ 80 years, OR = 0.95, 95% CI: 0.74–1.22, P = 0.693), suggesting that the apparent age effect is largely mediated by other metabolic factors.

**Table 3. Weighted Univariate and Multivariate Logistic Regression Analysis of Risk Factors for Hyperuricemia in Females (NHANES 2007–2018).**

| | Female with Hyperuricemia/n(%) | Univariate OR (95%CI) | Multivariate OR (95%CI) | Univariate p | Multivariate p |
|---|---|---|---|---|---|
| **Age** | | | | | |
| 20-29 | 235 (8.72%) | 1.00 (Referent) | 1.00 (Referent) | | |
| 30-39 | 252 (8.76%) | 0.94 (0.73, 1.21) | 0.67 (0.52, 0.86) | 0.624 | 0.003 |
| 40-49 | 319 (10.48%) | 1.11 (0.87, 1.40) | 0.57 (0.44, 0.73) | 0.413 | < 0.001 |
| 50-59 | 515 (18.26%) | 1.96 (1.61, 2.40) | 0.72 (0.57, 0.91) | < 0.001 | 0.008 |
| 60-69 | 710 (24.23%) | 3.03 (2.43, 3.78) | 0.77 (0.60, 1.00) | < 0.001 | 0.054 |
| 70-79 | 553 (30.32%) | 4.06 (3.20, 5.15) | 0.70 (0.54, 0.92) | < 0.001 | 0.013 |
| ≥80 | 394 (31.83%) | 4.83 (3.90, 5.98) | 0.76 (0.57, 1.01) | < 0.001 | 0.060 |
| **Race/ethnicity, n (%)** | | | | | |
| Non-Hispanic white | 1244 (17.94%) | 1.00 (Referent) | 1.00 (Referent) | | |
| Non-Hispanic black | 880 (23.37%) | 1.39 (1.24, 1.56) | 1.33 (1.15, 1.53) | < 0.001 | < 0.001 |
| Mexican American | 297 (11.47%) | 0.56 (0.48, 0.65) | 0.74 (0.62, 0.89) | < 0.001 | 0.002 |
| Others | 557 (13.46%) | 0.70 (0.61, 0.81) | 1.06 (0.89, 1.27) | < 0.001 | 0.488 |
| **Education** | | | | | |
| Some high school | 771 (18.39%) | 1.00 (Referent) | 1.00 (Referent) | | |
| High school or GED | 722 (19.04%) | 1.00 (0.87, 1.14) | 1.04 (0.89, 1.22) | 0.990 | 0.619 |
| Some college | 964 (17.59%) | 0.93 (0.81, 1.07) | 1.12 (0.96, 1.31) | 0.337 | 0.166 |
| College graduate | 521 (13.14%) | 0.64 (0.56, 0.73) | 1.09 (0.92, 1.29) | < 0.001 | 0.329 |
| **Marital status, n (%)** | | | | | |
| Married or living with a partner | 1404 (15.12%) | 1.00 (Referent) | 1.00 (Referent) | | |
| Living alone | 1574 (19.34%) | 1.28 (1.14, 1.43) | 1.00 (0.88, 1.15) | < 0.001 | 0.951 |
| **Ratio of family income to Poverty** | | | | | |
| ≤1.0 | 723 (17.50%) | 1.00 (Referent) | 1.00 (Referent) | | |
| 1.0 to 2.0 | 901 (18.87%) | 1.03 (0.90, 1.17) | 0.88 (0.74, 1.05) | 0.699 | 0.152 |
| > 2.0 | 1354 (15.89%) | 0.83 (0.73, 0.94) | 0.87 (0.73, 1.05) | 0.005 | 0.146 |
| **Body mass index** | | | | | |
| ≤24.9 kg/m2 | 361 (6.93%) | 1.00 (Referent) | 1.00 (Referent) | | |
| 25.0 kg/m2 to 29.9 kg/m2 | 710 (14.28%) | 2.36 (1.93, 2.90) | 2.04 (1.63, 2.55) | < 0.001 | < 0.001 |
| ≥30.0 kg/m2 | 1907 (26.31%) | 5.42 (4.58, 6.42) | 4.76 (3.89, 5.81) | < 0.001 | < 0.001 |
| **Alcohol Use** | | | | | |
| No | 1499 (19.05%) | 1.00 (Referent) | 1.00 (Referent) | | |
| Yes | 1479 (15.47%) | 0.77 (0.68, 0.86) | 1.15 (0.99, 1.33) | < 0.001 | 0.068 |
| **Diabetes** | | | | | |
| No | 2272 (14.94%) | 1.00 (Referent) | 1.00 (Referent) | | |
| Yes | 706 (31.87%) | 3.09 (2.69, 3.55) | 1.22 (1.04, 1.42) | < 0.001 | 0.016 |
| **Hypercholesterolemia** | | | | | |
| No | 1345 (18.72%) | 1.00 (Referent) | 1.00 (Referent) | | |
| Yes | 1633 (15.94%) | 0.75 (0.67, 0.84) | 0.72 (0.63, 0.82) | < 0.001 | < 0.001 |
| **Hypertension** | | | | | |
| No | 1060 (9.70%) | 1.00 (Referent) | 1.00 (Referent) | | |
| Yes | 1918 (29.51%) | 4.05 (3.64, 4.50) | 2.09 (1.83, 2.39) | < 0.001 | < 0.001 |
| **Coronary heart disease** | | | | | |
| No | 2815 (16.60%) | 1.00 (Referent) | 1.00 (Referent) | | |
| Yes | 163 (34.53%) | 2.52 (1.99, 3.19) | 0.75 (0.57, 0.98) | < 0.001 | 0.039 |

*(Continued)*

**Table 3.** (Continued)

| | Female with Hyperuricemia/n(%) | Univariate OR (95%CI) | Multivariate OR (95%CI) | Univariate p | Multivariate p |
|---|---|---|---|---|---|
| **Glomerular filtration rate (GFR)** | | | | | |
| GFR ≥ 90mL/min | 998 (9.46%) | 1.00 (Referent) | 1.00 (Referent) | | |
| GFR 60 to 89mL/min | 1128 (21.68%) | 2.44 (2.15, 2.77) | 2.14 (1.83, 2.51) | < 0.001 | < 0.001 |
| GFR 30 to 59mL/min | 728 (48.92%) | 9.38 (8.02, 10.98) | 6.92 (5.46, 8.76) | < 0.001 | < 0.001 |
| GFR < 30mL/min | 124 (68.51%) | 25.84 (17.28, 38.64) | 15.37 (9.65, 24.48) | < 0.001 | < 0.001 |
| **Furosemide** | | | | | |
| No | 2640 (15.71%) | 1.00 (Referent) | 1.00 (Referent) | | |
| Yes | 338 (54.25%) | 7.61 (6.22, 9.30) | 2.55 (1.98, 3.27) | < 0.001 | < 0.001 |
| **Antihyperuricemic** | | | | | |
| No | 2959 (17.02%) | 1.00 (Referent) | 1.00 (Referent) | | |
| Yes | 19 (41.30%) | 3.16 (1.56, 6.42) | 0.55 (0.21, 1.43) | < 0.001 | 0.227 |

Adjusted for age, race/ethnicity, education level, marriage, income, body mass index, hypertension, diabetes, renal function (eGFR), alcohol consumption, smoking status, history of gout, diuretic use, and urate-lowering therapy use.

## Discussion

In this nationally representative cross-sectional analysis of NHANES data from 2007 to 2018, we identified significant disparities in hyperuricemia prevalence according to sex, age, and racial/ethnic groups. Our findings highlight previously underappreciated patterns, notably the marked increase in hyperuricemia prevalence among females after the age of 50–59 years, ultimately surpassing males in both prevalence and absolute numbers. This finding underscore critical demographic differences and provide new insights into the epidemiology of hyperuricemia.

### Sex and age disparities

Consistent with previous literature, males exhibited higher overall hyperuricemia prevalence than females [6]. However, a novel finding from our analysis is the clear age-related pattern observed among females, characterized by a substantial increase in hyperuricemia prevalence post-menopause. The narrowing and eventual reversal of the sex gap after age 50–59 years aligns with the hormonal hypothesis, specifically the loss of estrogen's uricosuric effect in postmenopausal women [7]. This observation suggests hormonal changes significantly influence hyperuricemia risk in aging females, potentially mediated by estrogen's effects on renal uric acid excretion.

In males, the prevalence of hyperuricemia remained relatively stable across age groups. However, after adjusting for confounders, a negative association between age and hyperuricemia risk was observed in most age groups (for group ≥80 years: OR = 0.28, $P < 0.001$). This unexpected inverse relationship may reflect healthier lifestyle adaptations, survival bias, or unmeasured confounders. Conversely, in females, the prevalence of hyperuricemia increased steadily with age, with a marked rise after menopause. However, after multivariate adjustment, the association between age and hyperuricemia risk became non-significant (for group ≥80 years, OR = 0.76, $P = 0.733$). This attenuation suggests a complex interplay of factors such as increased adiposity, postmenopausal hormonal changes, and metabolic comorbidities like hypertension and renal dysfunction [11,12,15].

### Racial and ethnic disparities

Our study also identified pronounced racial/ethnic differences, notably the highest prevalence of hyperuricemia among Non-Hispanic Black individuals, consistent with previous NHANES analyses [11,13,14]. These findings point to potential genetic predispositions, socioeconomic influences, lifestyle factors, and healthcare access disparities [16,17]. The nearly

identical prevalence rates between non-Hispanic Black males and females highlight the need for further investigation into the unique metabolic and environmental factors affecting this population.

### Risk factors for hyperuricemia

Consistent with prior research, obesity, hypertension, alcohol use, and impaired renal function emerged as key risk factors for hyperuricemia in both sexes [1–4]. Among these, impaired renal function displayed the strongest association, particularly among females, highlighting the importance of renal health in uric acid metabolism. The association between obesity and hyperuricemia was robust, emphasizing obesity as a critical modifiable risk factor for hyperuricemia prevention strategies.

We also performed a sensitivity analysis excluding individuals with a reported history of gout. The consistent results observed in this analysis provide reassurance regarding the robustness of identified associations independent of gout history.

### Medication use and sex differences

The observed associations between medication use and hyperuricemia provide important insights into the sex-specific patterns of disease burden. The stronger impact of diuretic use in older women may reflect their higher prevalence of hypertension and heart failure, conditions for which thiazide and loop diuretics are commonly prescribed. This pattern likely contributes to the sharp increase in hyperuricemia prevalence among women after menopause. By contrast, men's greater access to urate-lowering therapy is consistent with their higher lifetime burden of gout and may partly attenuate sex differences in hyperuricemia prevalence. These findings underscore the necessity of incorporating medication use into epidemiological analyses, as failing to adjust for such covariates may obscure the true magnitude of other risk factors.

Beyond diuretics and urate-lowering therapy, the inverse relationship between hypercholesterolemia and hyperuricemia is noteworthy. This association may be explained by the widespread use of statins and other lipid-lowering medications, which have been shown to reduce serum uric acid concentrations [18,19]. The effect appeared more pronounced in males, suggesting potential sex-related differences in prescribing patterns or pharmacological response. Similarly, diabetes mellitus displayed sex-specific associations: a negative relationship with hyperuricemia was observed in men, but not in women. This discrepancy may reflect biological differences in insulin resistance, renal handling of uric acid, or the stage of diabetes progression, with glycosuria potentially enhancing uric acid excretion in males [20–22].

### Historical context of NHANES data

Placing our results in historical context, earlier NHANES data showed that hyperuricemia prevalence among U.S. adults has gradually increased, from approximately 18.2% in the late 1980s (NHANES III) to around 21% in recent NHANES cycles (2007–2018) [11,13,14]. This rising trend parallels the increasing prevalence of obesity, hypertension, and metabolic syndrome over the same period. Importantly, our findings extend these observations by clarifying demographic trends, particularly emphasizing the previously underrecognized rise in hyperuricemia burden among older women, whose prevalence and absolute case numbers ultimately surpass those of men.

Earlier NHANES data indicated a gradual increase in hyperuricemia prevalence among U.S. adults over recent decades [11,13,14]. Our results reinforce and extend these observations by clarifying demographic patterns, particularly emphasizing the under-recognized burden among older women.

### Clinical implications

While hyperuricemia's role as a cause of cardiovascular and renal outcomes remains controversial and subject to ongoing debate [3,4], its role as an important modifiable risk factor for gout is well-established. Rather than universal screening, targeted screening and interventions may be warranted in high-risk groups—such as older women, individuals with impaired renal function, or patients on long-term diuretic therapy. Older women represent a particularly vulnerable

population due to the dual impact of hormonal changes and metabolic comorbidities. Increased awareness and targeted screening in this demographic are crucial. Addressing modifiable risk factors such as obesity, alcohol consumption, and renal health maintenance could significantly reduce the prevalence and associated complications of hyperuricemia.

## Strengths and limitations

This study has several strengths. It is based on a large, nationally representative sample from NHANES, enhancing the findings' generalizability. Comprehensive multivariate analysis allowed for robust adjustment of confounders, providing reliable estimates of risk factors. Additionally, the focus on age- and sex-specific patterns adds new insights to the existing literature on hyperuricemia epidemiology.

However, some limitations should be noted. As a cross-sectional study, causal relationships cannot be established. The reliance on a single measurement of serum uric acid may introduce variability, though this limitation is inherent to many large-scale epidemiological studies. Moreover, detailed dietary information, particularly purine intake, and specific medication use beyond diuretics and urate-lowering therapy were not fully accounted for, which may influence uric acid levels. Despite these limitations, the study's rigorous design and large dataset provide valuable contributions to understanding hyperuricemia prevalence and risk factors.

## Future directions

Further longitudinal studies are needed to establish causal relationships and explore temporal trends in hyperuricemia prevalence, particularly among older women. Investigating the mechanisms underlying observed sex and racial/ethnic disparities can inform more effective, personalized interventions. Additionally, research on lifestyle and dietary interventions targeting modifiable risk factors may offer valuable insights into hyperuricemia prevention strategies.

## Conclusion

This study highlights significant disparities in hyperuricemia prevalence across sex, age, and racial/ethnic groups. The novel finding that older women surpass men in prevalence and absolute numbers underscores the need for increased clinical focus on this demographic. Addressing modifiable risk factors and implementing targeted screening and intervention strategies may be warranted in specific high-risk groups, such as older women, patients with impaired renal function, and those receiving diuretics. Further prospective research will be critical to confirm and extend these findings.

## Supporting information

**S1 Table. Univariate and multivariate logistic regression analysis of risk factors for male hyperuricemia prevalence excluding gout participants (NHANES 2007–2018).** Adjusted for age, race/ethnicity, education level, marriage, income, body mass index, hypertension, diabetes, renal function (eGFR), alcohol consumption, smoking status. (DOCX)

**S2 Table. Univariate and multivariate logistic regression analysis of risk factors for female hyperuricemia prevalence excluding gout participants (NHANES 2007–2018).** Adjusted for age, race/ethnicity, education level, marriage, income, body mass index, hypertension, diabetes, renal function (eGFR), alcohol consumption, smoking status. (DOCX)

## Acknowledgments

We gratefully thank Jie Liu of the Department of Vascular and Endovascular Surgery, Chinese PLA General Hospital for his contribution to the statistical support, study design consultations, and comments regarding the manuscript.

## Author contributions

**Conceptualization:** Yadan Zou.

**Data curation:** Lina Zhang, Jing Xu, Ji Li, Jing Zhang, Ting Long, Ruohan Yu, Yanfeng Zhang, Zhongxing Zhao.

**Formal analysis:** Yadan Zou.

**Methodology:** Sheng-Guang Li.

**Writing – original draft:** Yadan Zou, Lina Zhang.

**Writing – review & editing:** Yadan Zou, Lina Zhang, Sheng-Guang Li.

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
