## [Decision Letter · Decision Letter 0]

7 May 2025

Sex, Age, and Racial/Ethnic Disparities in Hyperuricemia Prevalence and Risk Factors Among U.S. Adults: An Analysis of NHANES 2007–2018 Data

PLOS ONE

Dear Dr. Li,

Thank you for submitting your manuscript to PLOS ONE. After careful consideration, we feel that it has merit but does not fully meet PLOS ONE’s publication criteria as it currently stands. Therefore, we invite you to submit a revised version of the manuscript that addresses the points raised during the review process.

The review comments can be found at the end of this email. Once you have re-submitted your manuscript, it will be re-reviewed before any final decision on possible publication is made. However, I cannot assure guarantee that the manuscript will be accepted after revision.

We look forward to receiving your revised manuscript.

Kind regards,

Toshiki Maeda

Academic Editor

PLOS ONE

Journal Requirements:

Additional Editor Comments:

**Comment**

Zou et al. evaluated the prevalence of hyperuricemia among U.S. adults by sex, age, and racial/ethnic groups and identified common and sex-specific risk factors using NHANES. As a result, they found significant disparities in hyperuricemia prevalence by sex, age, and race/ethnicity among U.S. adults. They also pointed out that females surpass males in prevalence and case numbers after age 50–59, which warrants greater clinical focus on older women. Although the findings are attractive and significant, there are some methodological concerns.

**Major comments**

Is hyperuricemia solely hazardous? Hyperuricemia may be related to cardiovascular events, but the causal relationship is still ambiguous. I could not understand why the author underlined that targeted screening and interventions are essential for hyperuricemia.Gout is caused by the precipitation of uric acid, which is derived from hyperuricemia. The relationship between gout and hyperuricemia is apparent, so there is no need to elaborate on it further.Did you consider medication use? The discrepancy in sex may be attributed to differences in medication use between the sexes, and you should discuss this further.

**Minor comments**

Please only suggest the results and avoid adding interpretation in the Results section. For example, “…, indicating that obesity is one of the most important modifiable risk factors.” in BMI. Such interpretation should be in the Discussion section.Typo: “base on the clinical guidelines….” Please use the uppercase letters. Additionally, please add the reference to the definition of hyperuricemia (“Hyperuricemia Identification” in “Data Collection and Variable Definitions”).I highly recommend using English editing by native.

Reviewers' comments:

Reviewer's Responses to Questions

**Comments to the Author**

1. Is the manuscript technically sound, and do the data support the conclusions?

Reviewer #1: Partly

Reviewer #2: Partly

2. Has the statistical analysis been performed appropriately and rigorously?

Reviewer #1: Yes

Reviewer #2: Yes

3. Have the authors made all data underlying the findings in their manuscript fully available?

Reviewer #1: Yes

Reviewer #2: Yes

4. Is the manuscript presented in an intelligible fashion and written in standard English?

Reviewer #1: Yes

Reviewer #2: Yes

Reviewer #1: Hello

This is a good article and it explores an important topic.

It seems that more studies are needed in this field in the future to be able to state with certainty the impact of exposure to such toxic substances.

Good luck.

Reviewer #2: This paper analyses NHANES 2007–2018 data to examine sex, age, and racial/ethnic disparities in hyperuricemia among U.S. adults. The authors report that hyperuricemia prevalence rises sharply among females after midlife, eventually surpassing males, and identify key risk factors including obesity, hypertension, alcohol use, and renal dysfunction. The findings highlight the need for targeted prevention strategies across demographic groups.

I have the following comments:

1. Consider the issue of reverse causality: Hyperuricemia is a precursor condition to gout, as the authors also acknowledge in the introduction. Including “history of gout” as a covariate in the multivariable regression model may introduce reverse causality bias. I recommend conducting a sensitivity analysis excluding individuals with a history of gout to better assess the risk factors for hyperuricemia.

2. Clarify laboratory methods: Please provide a detailed description of how serum uric acid was measured in NHANES, including the assay method, equipment, and quality control procedures if available. This will improve reproducibility and methodological transparency.

3. Definition of hyperuricemia: Specify clearly which guideline was used to define hyperuricemia (e.g., ACR, EULAR, or another standard). Please cite the source and briefly discuss the clinical rationale for the chosen cut-offs.

4. Provide more detail on covariates: Describe key variables more clearly. For example, what were the specific categories of education? Were these self-reported during interviews, or collected through another method (e.g., online forms or medical examination)?

5. Expand on statistical analysis methods:

5.1: Specify which statistical software and packages were used.

5.2: Confirm whether survey weights were applied during analysis, given the complex NHANES sampling design.

5.3: Provide more information about how covariates were selected for inclusion in the multivariable models (e.g., based on prior literature, significance in univariable models, directed acyclic graphs, etc.).

5.4: Indicate whether regression diagnostic tests (such as checking for multicollinearity, goodness-of-fit, or influential observations) were conducted.

6. Contextualize findings with previous NHANES data: It would strengthen the discussion to mention hyperuricemia prevalence from prior NHANES cycles (e.g., 1988–1994, 1999–2000) and highlight any trends over time. This would help situate the current findings within the broader epidemiological context.

**Do you want your identity to be public for this peer review?** For information about this choice, including consent withdrawal, please see our Privacy Policy

Reviewer #1: No

Reviewer #2: No

---

## [Author Response · Author response to Decision Letter 1]

3 Nov 2025

Dear Editor,

We thank you and the reviewers for your careful evaluation of our manuscript, “Sex, Age, and Racial/Ethnic Disparities in Hyperuricemia Prevalence and Risk Factors Among U.S. Adults: An Analysis of NHANES 2007–2018 Data.” We appreciate all the comments and suggestions, which have helped us improve the clarity and quality of the paper. We have revised the manuscript accordingly. Below, we provide a point-by-point response to each comment raised by the editor and reviewers. For clarity, the original comments are quoted in italics (indented), and our responses follow. The corresponding revisions in the manuscript have been highlighted in red. A clean version of the revised manuscript has also been uploaded.

Editor’s Comments

Comment 1 (Editor): “Is hyperuricemia solely hazardous? Hyperuricemia may be related to cardiovascular events, but the causal relationship is still ambiguous. I could not understand why the author underlined that targeted screening and interventions are essential for hyperuricemia.”

Response:

Thank you very much for raising this insightful question, which gives us the opportunity to clarify further the rationale behind emphasizing “targeted screening and intervention for hyperuricemia (HUA)” in our manuscript. We fully acknowledge that the causal relationship between hyperuricemia and cardiovascular events remains uncertain, a viewpoint supported by several recent Mendelian randomization studies and prospective cohort investigations1,2.

It was not our intention to imply that hyperuricemia is invariably harmful across all populations, nor to advocate for universal intervention in the general public. Our emphasis on “targeted screening and intervention” stems from the fact that, even in the absence of definitive causal evidence, extensive epidemiological studies have consistently shown that hyperuricemia is associated with an increased risk of gout3, progression of chronic kidney disease4, metabolic abnormalities, and incident cardiovascular events5. Moreover, among individuals with established cardiometabolic or renal risk factors—such as hypertension, obesity, or chronic kidney disease—elevated uric acid often serves as an early and modifiable biomarker6,7.

Therefore, by “targeted screening and intervention,” we refer to strategies focused on specific high-risk groups—such as those with recurrent gout attacks, impaired renal function, or multiple metabolic risk factors. In these populations, timely monitoring and evidence-based intervention when clinically indicated may help reduce the incidence of complications. We have expanded on the scope and rationale of this approach in the Discussion section to avoid overgeneralizing our recommendations to the general population8.

We have revised the Conclusions(L503-L506) to state that targeted screening and interventions may be warranted in specific high-risk groups, rather than implying it is essential for everyone. These changes ensure a balanced discussion that does not overstate the hazards of hyperuricemia while still highlighting the public health relevance of our findings.

Comment 2 (Editor): “Gout is caused by the precipitation of uric acid, which is derived from hyperuricemia. The relationship between gout and hyperuricemia is apparent, so there is no need to elaborate on it further.”

Response:

We agree and have streamlined the manuscript to avoid over-elaborating on the well-known relationship between hyperuricemia and gout. In the Introduction, we initially provided background on gout as a consequence of hyperuricemia; we have now condensed this to a brief statement for context only. Any redundant or lengthy explanation of gout’s relationship to hyperuricemia has been removed. This revision keeps the focus on our study’s objectives and findings, without reiterating established knowledge.

Comment 3 (Editor): “Did you consider medication use? The discrepancy in sex may be attributed to differences in medication use between the sexes, and you should discuss this further.”

Response:

Thank you for raising this critical point. In the revised manuscript, we have gone beyond discussion and included an analysis of two classes of medications available in NHANES that directly affect serum uric acid levels: diuretics and urate-lowering therapy (ULT).

Our weighted regression models demonstrate that the use of diuretics (e.g., furosemide) was strongly associated with increased odds of hyperuricemia in both sexes. After full adjustment, the odds ratio for males was 2.67 (95% CI: 1.97–3.61) and for females was 2.55 (95% CI: 1.98–3.27). Conversely, the use of urate-lowering medications (e.g., allopurinol, febuxostat) was significantly associated with a reduced risk of hyperuricemia. The adjusted odds ratio was 0.57 (95% CI: 0.35–0.95) in males and 0.55 (95% CI: 0.21–1.43) in females, thou-gh the result for females did not reach statistical significance, likely due to limited sample size. These findings confirm the expected biological effects of these medications and highlight their role as essential covariates.

We also note sex-related differences: diuretic use was more common in older women (likely due to treatment of hypertension and heart failure), which may contribute to the higher prevalence of hyperuricemia observed in females after midlife. On the other hand, men had a higher prevalence of urate-lowering therapy use, reflecting their higher lifetime burden of gout, which may partly offset hyperuricemia prevalence in males.

In the revised Abstract (L78–82), Methods (L244–240), Results (L333–339), and Discussion (L412–423), we explicitly describe the findings on medication use and interpret them as contributing to the observed sex disparity in hyperuricemia prevalence. With the inclusion of medication variables in the multivariable models (see Tables 2 and 3), estimates for other risk factors became more robust. This addition strengthens the manuscript by moving beyond speculation and providing direct evidence of the impact of medication use on serum uric acid levels.

Table 2. Weighted Univariate and Multivariate Logistic Regression Analysis of Risk Factors for Hyperuricemia in Males (NHANES 2007–2018)

Adjusted for age, race/ethnicity, education level, marriage, income, body mass index, hypertension, diabetes, renal function (eGFR), alcohol consumption, smoking status, history of gout, diuretic use, and urate-lowering therapy use.

Table 3. Weighted Univariate and Multivariate Logistic Regression Analysis of Risk Factors for Hyperuricemia in Females (NHANES 2007–2018)

Adjusted for age, race/ethnicity, education level, marriage, income, body mass index, hypertension, diabetes, renal function (eGFR), alcohol consumption, smoking status, history of gout, diuretic use, and urate-lowering therapy use.

Comment 4 (Editor): “Please only suggest the results and avoid adding interpretation in the Results section. For example, ‘…, indicating that obesity is one of the most important modifiable risk factors.’ in BMI. Such interpretation should be in the Discussion section.”

Response: We have revised the Results section to remove any interpretative statements, ensuring it focuses strictly on the empirical findings. Specifically, we deleted the phrase “indicating that obesity is one of the most important modifiable risk factors” from the Results section where we report the association with BMI.

We relocated this interpretative insight to the Discussion(L410-L412) section, where we discuss the implications of the strong association between obesity and hyperuricemia. The revised manuscript is as followed:

“The association between obesity and hyperuricemia was robust, emphasizing obesity as a critical modifiable risk factor for hyperuricemia prevention strategies”.

Throughout the Results, we now present the data without commentary, and we reserve interpretation, context, and implications for the Discussion. This change aligns the manuscript with the journal’s guidelines and clearly separates results from interpretation.

Comment 5 (Editor): “Typo: ‘based on the clinical guidelines….’ Please use the uppercase letters. Additionally, please add the reference to the definition of hyperuricemia (‘Hyperuricemia Identification’ in ‘Data Collection and Variable Definitions’).”

Response:

Thank you very much for your careful review and thoughtful corrections regarding these details. We have made the following revisions based on your suggestions:

1) Correction of Spelling and Formatting

In the “Data Collection and Variable Definitions” section, the phrase “based on the clinical guidelines” has been revised to “Based on the clinical guidelines,” with the initial letter capitalised in accordance with English writing conventions. We have also conducted a thorough review of the entire manuscript to ensure that similar spelling and capitalization issues have been uniformly corrected to maintain consistency in formatting.

2) Addition of Reference for the Definition of Hyperuricemia

In the “Hyperuricemia Identification” subsection, we have supplemented the definition of hyperuricemia with an authoritative reference to allow readers to access the original source. The revised manuscript is as followed:

“Hyperuricemia was defined as a serum uric acid (SUA) concentration >7.0 mg/dL (>416 μmol/L) in males and >6.0 mg/dL (>357 μmol/L) in females. These thresholds were based on the 2018 European Alliance of Associations for Rheumatology (EULAR) evidence-based recommendations for the diagnosis of gout7 and the 2020 American College of Rheumatology (ACR) guideline for the management of gout9”

Comment 6 (Editor): “I highly recommend using English editing by native.”

Response:

We have thoroughly revised the manuscript for English language and style. A native English-speaking colleague (and professional editor) assisted in copyediting the text. We corrected grammatical errors, improved sentence clarity, and eliminated awkward phrasing. The overall readability of the manuscript has been significantly improved. Additionally, we took this opportunity to ensure that the formatting and style conform to PLOS ONE guidelines. For example, we adjusted the manuscript to meet the journal’s formatting requirements for headings, figures, and references. We trust that the language and presentation of the paper now meet the high standards expected by the journal.

Reviewer #1’s Comments

Comment (Reviewer 1): “Hello. This is a good article, and it explores an important topic. It seems that more studies are needed in this field in the future to be able to state with certainty the impact of exposure to such toxic substances. Good luck.”

Response:

We thank Reviewer #1 for the positive feedback and encouragement. We are pleased that the importance of the topic was acknowledged. We agree that further studies (especially longitudinal and interventional research) are needed to conclusively determine the causal impacts of chronic hyperuricemia on health outcomes. In the revised Discussion, we have noted that our cross-sectional findings highlight associations and disparities that warrant prospective investigation. We also emphasize in the Conclusion that continued research will be important to confirm and extend our findings. We appreciate the reviewer’s insight and have kept this perspective in mind, indicating that while our study adds to the knowledge in this field, establishing causal effects will require future investigations.

Reviewer #2’s Comments

Comment 1 (Reviewer 2): “Consider the issue of reverse causality: Hyperuricemia is a precursor condition to gout, as the authors also acknowledge in the introduction. Including ‘history of gout’ as a covariate in the multivariable regression model may introduce reverse causality bias. I recommend conducting a sensitivity analysis excluding individuals with a history of gout to better assess the risk factors for hyperuricemia.”

Response:

We sincerely appreciate this valuable suggestion regarding potential reverse causality. We have carefully addressed this concern through a sensitivity analysis, and the key revisions are as follows:

1) Sensitivity analysis:

As recommended, we conducted a sensitivity analysis by excluding all participants with a self-reported history of gout and re-ran the multivariable logistic regression models for hyperuricemia.

2) Summary of Findings:

The results from this analysis remained highly consistent with those from the primary model. Specifically:

o No material changes were observed in the significance or direction of any key risk factors.

o For instance, in females, the adjusted odds ratio for obesity changed only marginally from 4.29 (full sample) to 4.44 (gout-excluded sample), with fully overlapping 95% confidence intervals.

o Similar stability was noted across all sex and racial/ethnic subgroups.

o 3) Manuscript Additions:

o A statement has been added to the Results section highlighting the consistency of the sensitivity analysis results.

o Two new supplementary tables (Table S1 for males and Table S2 for females) have been included, presenting adjusted prevalence and odds ratios after excluding participants with gout.

o 4) Interpretation:

The minimal effect of excluding gout patients suggests that reverse causality did not substantially bias our original estimates. This reinforces the robustness of the identified risk factors for hyperuricemia, even after accounting for potential behavioral or pharmacological modifications among gout patients.

5) Discussion Update:

We have added a comment in the Discussion noting that the associations reported are robust to the exclusion of individuals with a history of gout.

Table S1. Univariate and Multivariate Logistic Regression Analysis of Risk Factors for Male Hyperuricemia Prevalence excluding gout participants (NHANES 2007–2018)

Table S2. Univariate and Multivariate Logistic Regression Analysis of Risk Factors for Female Hyperuricemia Prevalence excluding gout participants (NHANES 2007–2018)

Adjusted for age, race/ethnicity, education level, marriage, income, body mass index, hypertension, diabetes, renal function (eGFR), alcohol consumption, smoking status.

Comment 2 (Reviewer 2): “Clarify laboratory methods: Please provide a detailed description of how serum uric acid was measured in NHANES, including the assay method, equipment, and quality control procedures if available. This will improve reproducibility and methodological transparency.”

Response: Thank you for your suggestions. We have expanded the Methods section to include a more detailed description of the laboratory measurement of serum uric acid in NHANES.

In the revised “Laboratory Measurements” subsection(L174-L181), we now state that serum uric acid was measured enzymatically using a uricase-based method. We specify the instruments used by NHANES: in the 2007–2008 cycle, uric acid was measured with a Beckman Synchron LX20 auto-analyzer, and from 2008 onward NHANES utilized the Beckman Coulter UniCel DxC800 analyzer. The assay involved a timed endpoint method in which uricase catalyzes the oxidation of uric acid to allantoin and hydrogen peroxide; the reaction is coupled to a colorimetric change that is measured photometrically, and the change in absorbance is directly proportional to the uric acid concentration in the sample.

We also note that NHANES employed rigorous quality control procedures: all laboratory testing followed standardized protocols with regular calibration, and quality control samples were analyzed to ensure accuracy and consistency. NHANES laboratory data are subject to external quality assurance through the Centers for Disease Control and Prevention’s protocols. These details have been added to the Methods for transparency. We believe that providing this information will help with reproducibility and give readers confidence in the reliability of the laboratory data.

Comment 3 (Reviewer 2): “Definition of hyperuricemia: Specify clearly which guideline was used to define hyperuricemia (e.g., ACR, EULAR, or another standard). Please cite the source and briefly discuss the clinical rationale for the chosen cut-offs.”

Responses:

Thank you very much for the constructive feedback. In the revised manuscript, we have provid

---

## [Editor Report · Decision Letter 1]

13 Nov 2025

Sex, Age, and Racial/Ethnic Disparities in Hyperuricemia Prevalence and Risk Factors Among U.S. Adults: An Analysis of NHANES 2007–2018 Data

PONE-D-25-05451R1

Dear Dr. Li,

We’re pleased to inform you that your manuscript has been judged scientifically suitable for publication and will be formally accepted for publication once it meets all outstanding technical requirements.

Kind regards,

Toshiki Maeda

Academic Editor

PLOS ONE

Additional Editor Comments (optional):

The concerns are addressed satisfactory.
---

## [Editor Report · Acceptance letter]

PONE-D-25-05451R1

PLOS One

Dear Dr. Li,

I'm pleased to inform you that your manuscript has been deemed suitable for publication in PLOS One. Congratulations! Your manuscript is now being handed over to our production team.

Kind regards,

on behalf of

Dr. Toshiki Maeda

Academic Editor

PLOS One